# Postconditioning by Delayed Administration of Ciclosporin A: Implication for Donation after Circulatory Death (DCD)

**DOI:** 10.3390/ijms232112858

**Published:** 2022-10-25

**Authors:** René Ferrera, Marie Védère, Megane Lo-Grasso, Lionel Augeul, Christophe Chouabe, Gabriel Bidaux, Delphine Baetz

**Affiliations:** University of Lyon, CARMEN Laboratory, INSERM, INRAE, Université Claude Bernard Lyon 1, 69500 Lyon, France

**Keywords:** donation after circulatory death (DCD), heart transplantation, ciclosporin A-postconditioning–ischemia reperfusion injury (IRI)-delayed reperfusion-mitochondria-permeability transition pore

## Abstract

Heart transplantation is facing a shortage of grafts. Donation after Circulatory Death (DCD) would constitute a new potential of available organs. In the present work, we aimed to evaluate whether Postconditioning (ischemic or with ciclosporin-A (CsA)) could reduce ischemia-reperfusion injury in a cardiac arrest model when applied at the start of reperfusion or after a delay. An isolated rat heart model was used as a model of DCD. Hearts were submitted to a cardiac arrest of 40 min of global warm ischemia (37 °C) followed by 3 h of 4 °C-cold preservation, then 60 min reperfusion. Hearts were randomly allocated into the following groups: control, ischemic postconditioning (POST, consisting of two episodes each of 30 s ischemia and 30 s reperfusion at the onset of reperfusion), and CsA group (CsA was perfused at 250 nM for 10 min at reperfusion). In respective subgroups, POST and CsA were applied after a delay of 3, 10, and 20 min. Necrosis was lower in CsA and POST versus controls (*p* < 0.01) whereas heart functions were improved (*p* < 0.01). However, while the POST lost its efficacy if delayed beyond 3 min of reperfusion, CsA treatment surprisingly showed a reduction of necrosis even if applied after a delay of 3 and 10 min of reperfusion (*p* < 0.01). This cardioprotection by delayed CsA application correlated with better functional recovery and higher mitochondrial respiratory index. Furthermore, calcium overload necessary to induce mitochondrial permeability transition pore (MPTP) opening was similar in all cardioprotection groups, suggesting a crucial role of MPTP in this delayed protection of DCD hearts.

## 1. Introduction

The main problem encountered in heart transplantation has been the shortage of grafts for many years. While the number of available organs remains stable for years, the number of patients on the waiting list continues to grow. It is, therefore, necessary to search for new sources of grafts. Organs from patients with Donation after Circulatory Death (DCD) could become a potentially significant supply of grafts, allowing them to break through the glass ceiling [1]. Even if, for several years, clinical studies confirmed the relevance of such approaches [2], DCD hearts remain damaging grafts due to the unavoidable initial ischemic suffering. As it is not allowed to intervene on DCD donors for ethical reasons (period of “no touch”), the other means of protecting the graft are after removal, either during transport of the organ or at the time of reperfusion [3]. Indeed, it is well-admitted that reperfusion following myocardial ischemia induces a new wave of lethal damage due to calcium overload, accelerated production of reactive oxygen species, and mitochondrial dysfunctions [4]. Cardioprotective strategies have been evaluated to protect the cardiac graft during reperfusion: Longnus’ group emphasized the key role of mitochondria, using three postconditioning (POST) strategies (mild hypothermia, mechanical postconditioning, and hypoxia) in an isolated rat heart model of DCD [5]. Akande et al. confirmed the damaging impact of DCD hearts on mitochondrial function [6]. These studies strongly suggest that mitochondria constitute a target of choice for the implementation of postconditioning strategies for DCD at the time of reperfusion.

However, despite its evident potential, one limitation of POST is the necessity to be achieved right at the beginning of reperfusion. Indeed, Kin et al. demonstrated that the benefit of mechanical ischemic POST is lost when it is delayed by 1 min after the onset of reperfusion in a rat model, suggesting a limited time window for POST efficacy at the onset of reperfusion [7]. In a rabbit model, Yang et al. confirmed that ischemic POST must occur within the first minutes of reperfusion to be protective [8]. A similar observation was obtained with an adenosine receptor agonist that lost its cardioprotective effect with delayed administration at reperfusion [9]. These works, performed on small animals, suggested that delaying the intervention after the onset of reperfusion would abolish the potentially protective effects of POST. One of the primary reasons for the ischemic failure of POST in small animals and probably in larger species, including humans, may be the delay in the intervention of the maneuver. This narrow therapeutic time window may limit the clinical use to patients in whom POST cannot be immediately and accurately performed. On the other hand, experimental evidence supports the mitochondrial permeability transition pore (MPTP) as a key mediator of cardioprotection and limitation of lethal myocardial reperfusion injury [10].

The present study questioned whether the direct inhibition of the mitochondrial permeability transition pore (MPTP) with ciclosporin A (CsA) induced POST remained cardioprotective and protected against lethal reperfusion injury even after a delayed intervention. The answer to this question could be useful to better understand the post-ischemic management of grafts from DCD patients.

## 2. Results

### 2.1. Delayed CsA Treatment Improved Myocardial Recovery

Baselines RPP (before ischemia) were comparable in all groups averaging 26,721 ± 164 mmHg/min. After ischemic periods, reperfusion-induced recovery of mean RPP (average between 30- and 60-min reperfusion) was impaired in all ischemic groups (*p* < 0.0001 versus baseline values). As expected, mean RPP measurements were better preserved in POST-0 and CsA-0 groups reaching respectively 11,501 ± 1507 and 9084 ± 341 mmHg/min (*p* < 0.001 versus control), (See Figure 1). In the groups with a delay of 3 min (POST-3 and CSA-3), the mean RPP value was also significantly enhanced compared to the control but similar to POST-0 and CsA-0. Intriguingly, this protective effect on myocardial function was abolished in POST-10 and POST-20 but maintained in CsA-10. Similarly, mean dP/dt max and min, PGVD and RPP were significantly improved in the CsA group up to 10 min of intervention compared to control values (*p* < 0.001). Conversely, POST lost its efficiency after 10 min of delay. (See Table 1).

### 2.2. Delayed CsA Treatment Reduced Myocardial Infarction

Ischemia reperfusion induced significant necrosis (LDH release) in all groups as compared with the sham group (*p* < 0.001) (Figure 2a). In line with the existing data, LDH release was significantly lower in POST-0 and CsA-0 versus the control group. LDH release was also reduced in delayed groups POST-3 (*p* = 0.033 versus control group), and in all CsA groups (CsA-3, CsA-10 and CsA-20). (Figure 2a).

TTC staining revealed that infarct sizes were significantly reduced in all CsA groups, until 10 min of delayed intervention averaging 20% ± 1 for CsA-0 (*p* = 0.0077), 17% ± 3 for CsA-3 (*p* = 0.0052) and 19% ± 2 for CsA-10 (*p* = 0.0067) compared with control values (34% ± 2). Conversely, delayed POST above 3 min lost efficacy (Figure 2b).

### 2.3. Delayed CsA Treatment Diminished MPTP Opening

The effect of CsA and POST on mitochondrial permeability transition pore (MPTP) opening was indirectly determined by measuring by CRC and is shown in Figure 3. In the sham group, during the presence of complex II substrates, the amount of Ca^2+^ required to open the MPTP averaged 831 ± 81 nmol/mg proteins. CsA treatment led to a higher CRC in all groups, compared with control (*p* < 0.001 for CsA-3 and CsA-10, and *p* = 0.038 for CsA-20), whereas MPTP inhibition was abolished in delayed POST 10 and 20. Similar results were obtained using complex I substrates (see Appendix A).

### 2.4. Mitochondrial Oxygen Consumption Was Enhanced with Delayed CsA Treatment

The mitochondrial oxygen consumption is shown in Table 2. In the sham group, the RCI (state III/State IV) averaged 5.23 ± 0.17. Ischemia reperfusion significantly reduced state 3 respiration and RCI when using complex I substrates, reaching 1.63 ± 0.04 in the control group (*p* = 0.0029 vs. sham). In POST-0 and CsA-0 groups, RCI was significantly higher than in control hearts, averaging respectively 2.10 ± 0.09 and 2.39 ± 0.30 (*p* = 0.044 and *p* = 0.0067 respectively vs. control).

In delayed POST, RCI was abolished and comparable to controls, whereas it remained significantly higher for delayed CsA-3 and CsA-10 (*p* < 0.05 vs. control).

### 2.5. Delayed CsA Reduced H_2_O_2_ Generation by Mitochondria

As ROS production depends on the level of mitochondrial stimulation, we defined an index of H_2_O_2_ production by dividing the latter with State III. The effect of CsA and POST on mitochondrial H_2_O_2_ index (H_2_O_2_ production/State III) is shown in Figure 4. In the sham group and the presence of complex I substrates (pyruvate, glutamate, malate), the basal H_2_O_2_ index averaged 0.240 ± 0.04. The H_2_O_2_ index was increased by the addition of rotenone and antimycin A. As expected, in the control group, H_2_O_2_ indexes were significantly increased in all conditions of stimulation (*p* < 0.001 versus sham). ROS production was reduced in POST 0 and POST-3 in all conditions (*p* < 0.001 versus control) and returned around control values after 10 min. However, ROS production was blunted in all CsA groups, especially under antimycin induction (*p* < 0.001 for CsA-0, 3, and 10, *p* = 0.043 for CsA 20 versus control). Similar results were obtained with substrates of complex II (see Appendix A).

## 3. Discussion

Using the Langendorff model of an isolated ischemic heart, we demonstrated for the first time that it is possible to delay the myocardial protection intervention by using CsA up to 10–20 min after the onset of reperfusion. This finding may be relevant for grafts submitted to warm ischemic insult, such as the Maastricht III heart graft harvested from DCD donors. Indeed, we evidenced a cardioprotective effect associated with reduced necrosis and higher myocardial function during reperfusion. Our data also suggest that better preservation of mitochondrial functions could be the mechanism responsible for this beneficial effect.

### 3.1. POST and CsA Gave Similar Protection

In the present study, we confirm first that both ischemic POST and CsA applied at the onset of reperfusion blunted the irreversible damage of ischemia-reperfusion. All functional parameters were improved in postconditioned or CsA-treated hearts. Interestingly, this protection observed on myocardial function during reperfusion has been detected as soon as 10 min after the start of reflow and was maintained throughout the reperfusion period.

Functional improvements were related to reduced necrosis (reduced infarct size and LDH release). Cell death resulting from ischemia-reperfusion may be attributed to several mechanisms, including MPTP opening causing necrosis and/or apoptosis [11]. MPTP opening is a critical and probably late event in cardiomyocyte death following ischemia at the onset of reperfusion. MPTP opening is controlled by several factors, including matrix Ca^2+^ accumulation and free radicals concentration changes [12]. 

Note that we did not observe an additive effect POST + CsA (see Appendix A). 

### 3.2. Consequence of Delayed Cardioprotective Interventions at Reperfusion: Place of CsA

The initial phase of reperfusion appears crucial for the pathogenesis of postischemic injury, and the delayed application of POST at reperfusion is a debated option. Indeed, several studies demonstrated that the protective effect of POST was lost when the intervention was delayed by only one minute after the onset of myocardial reperfusion [7,8,9]. Furthermore, on other organs such as the brain [13], lung [14], or intestines [15], the protective effect of POST failed when therapeutic interventions were delayed beyond 3 min. Moreover, the ischemic POST maneuver is not always simple to apply and requires interventions of occlusion–deocclusion, which can be deleterious on the already weakened artery. The present study partially supports this concept and confirms that, in our model, ischemic postconditioning is efficient but only when applied within 3 min at the onset of reperfusion.

Consequently, it seems that the therapeutic time window would be extremely narrow (in the case of ischemic postconditioning), which may limit the clinical translation to patients in whom reperfusion cannot be immediately and accurately established. Indeed, because of technical constraints such as using a thrombus aspiration procedure, POST cannot always be immediately performed at the beginning of reperfusion.

In the present study, we demonstrated that CsA could be successfully applied even 10 min after the onset of myocardial reperfusion. CsA-10 surprisingly induced significant protection without any major loss of efficacy compared to non-delayed POST. Interestingly, cell necrosis, assessed by LDH release, was also significantly attenuated after 20 min of delay (*p* < 0.05 between control and CsA-20, see Figure 2, panel a), suggesting that CsA may even be efficient after a long time of postischemic reperfusion. Furthermore, mitochondrial function (respiration and CRC) were well preserved in delayed CsA-10 (but not in POST-10), suggesting the key role of mitochondria in this delayed protection. Moreover, even if we have no precise explanation for this phenomenon, it appears that the production of ROS by the mitochondrial respiratory chain is lower in all the protected groups, whether under stimulation of complex I or complex II (see Figure 4 and Appendix A). We can assume that the better management of ROS by the mitochondria would be a key factor in cardioprotection and MPTP regulation, as suggested by Penna et al. [16]. However, in this study, we lack elements to support these assumptions concerning the role of ROS.

Ultimately, this study challenged the pre-established concept of the concise therapeutic window and suggested that it is possible to protect the heart graft even after the initiation of the reperfusion. Another study of our group had already indicated that the cardioprotective strategy named “low pressure reperfusion” (another form of postconditioning) might be applied with success even after a delay during reperfusion [17].

Therefore, the possibility of salvaging ischemic graft, even beyond the onset of reperfusion, remains an open avenue for translation to clinical studies. Ischemic postconditioning remains the method of choice to protect the ischemic heart graft at the time of reperfusion. However, the time window is very short and is a limitation. In addition, a series of clamping–unclamping of the aorta at the time of the transplantation would be difficult and possibly dangerous to implement. Postconditioning with ciclosporin A (CsA) counteracts these two constraints: indeed, CsA is simple to implement, and we demonstrated here that it might be protective, even if applied 10–20 min after the start of reperfusion. However, this result must be confirmed by other teams and in larger species.

## 4. Methods and Materials

### 4.1. Surgical Procedure

The investigation conformed to the *Guide for the Care and Use of Laboratory Animals* published by the US National Institute of Health (NIH Publication No. 85–23, revised 1996). Moreover, regarding regulation, our laboratory is authorized on behalf of the Direction of Veterinary Services. The following protocol was controlled and validated by the Ethics committee of the Claude Bernard University of Lyon.

Adult rats, all male (to avoid the influence of estrogen hormones) and 400–450 g (between 2 and 3 months old), were used. The animals were placed in an aesthetic chamber containing sevoflurane for induction. After the loss of consciousness, the animal is placed in dorsal recumbency with a breathing mask under sevoflurane (mac 4–5) for maintenance of anesthesia. Analgesia was performed with the α_2_ agonist xylazine (10 mg/kg). General deep anesthesia was controlled by the absence of reflexes (pedal, palpebral, corneal, pupillary light reflex) and lack of reaction to pain (plucking legs). Heparin (200 IU/kg) was injected into the femoral vein. Euthanasia was done by heart harvesting: hearts were removed and quickly installed (in less than a minute) on the Langendorff apparatus. Basal perfusion started immediately for 20 min using normothermic Krebs solution, and the heart was paced at a constant rate of 300 beats/min, as previously described [18].

Animals (n = 120) were allocated into 10 groups of 12 animals. See Figure 5. All hearts (except the shams) were submitted to 40 min of global warm ischemia at 37 °C (imitating a cardiac arrest), followed by 3 h of cold (4 °C) ischemic preservation in the Plegisol cardioplegic solution (imitating the period of hypothermic transportation of the graft), then followed by 60 min reperfusion. Animals were randomly assigned into one of the 10 following groups (n = 12/group):-Control group: myocardium was reperfused with Krebs following the ischemic insult.-POST-0 group (postconditioning without delay): full coronary reflow was performed for 30 s, followed by 30 s of global ischemia, repeated twice (2 min of total intervention) immediately at the beginning of reperfusion. Then, reperfusion was maintained until the end of the experiment.-CsA-0 group (cyclosporin A given at reperfusion without delay): the myocardium was reperfused with Krebs containing CsA at 250 nM during the first 10 min of reperfusion.-For delayed groups, POST and CsA interventions were deferred by 3, 10, or 20 min: groups POST-3, -10, -20 and CsA-3, -10, -20.

The sham group (n = 12) underwent no period of warm ischemia.

For CsA groups, CsA was dissolved in DMSO (vehicle). In the other groups (without CsA), the vehicle alone was added to the reperfusion medium.

### 4.2. Functional, Tissular, and Molecular Analysis

Two sets of experiments were performed:-Set I (10 groups) evaluated functional recovery and tissue necrosis after ischemia reperfusion.-Set II (10 groups) assessed mitochondrial functions: (1) oxygen consumption, (2) Ca^2+^-induced permeability transition pore opening, and (3) H_2_O_2_ production. Mitochondria were extracted from the whole heart (to reflect global ischemia).

#### 4.2.1. Functional Recovery

Heart rate (HR), left ventricular systolic pressure (LVSP), and left ventricular end-diastolic pressure (LVEDP) were measured using a latex balloon positioned into the left ventricle and inflated with water to exert a physiologic end-diastolic pressure of 5 mmHg. Rate-pressure product [RPP = (LVSP-LVEDP) · HR], maximum rate of rise of the LV pressure (dP/dt max) and maximum isovolumetric rate of relaxation (dP/dt min) were calculated. Coronary flow (CF) was measured by collecting the coronary effluent.

#### 4.2.2. Myocardial Necrosis

Cellular injury was evaluated by measurement of lactate dehydrogenase (LDH) release in the coronary effluent during the reperfusion period (Beckman Coulter kit, Galway, Ireland), as previously described [10]. Results were expressed in IU per ml and per mg of myocardial tissue (IU/mL/mg). Myocardial necrosis was also assessed using TTC staining. The total area of necrosis (AN) was then measured and expressed as a percentage of the total left ventricle (LV) area.

#### 4.2.3. Mitochondrial Oxygen Consumption

Oxygen consumption in freshly isolated mitochondria was measured at 25 °C with a Clark-electrode (Oroboros Oxygraph, Innsbruck, Austria). Mitochondria (250 µg proteins) were incubated in a respiratory buffer and pyruvate, malate, and glutamate (5 mmol/L each) were used as substrates to provide electrons to complex I. State 3 (200 µmol/L ADP addition) and state 4 (ADP limited) were assessed, and the Respiratory Control Index (RCI= State III/State IV) was calculated.

#### 4.2.4. Calcium Retention Capacity (CRC)

Calcium retention capacity (CRC) was defined as the amount of Ca^2+^ necessary to induce a massive Ca^2+^ release (MPTP opening) by isolated cardiac mitochondria. CRC is used as an indicator of the susceptibility of MPTP to Ca^2+^ overload. It was expressed as nmol CaCl_2_ per mg mitochondrial proteins. Extra-mitochondrial Ca^2+^ was estimated fluorometrically using 0.5 µM calcium green-5N (Molecular Probes™, Sigma Aldrich, L’Isle-d’Abeau Chesnes, France). Excitation and emission of calcium green wavelengths were fixed at 500 and 530 nm, respectively. Mitochondria (250 μg proteins) were incubated with 2 mL buffer, and CRC was assessed in the presence of 5 mM of respective substrates of complex I (pyruvate-glutamate-malate) or complex II (succinate).

A schematic diagram summarizing the principle of CRC measurement is presented in Appendix A to better understand the principle of CRC measurement.

#### 4.2.5. Mitochondrial H_2_O_2_ Production

The quantification of extra-mitochondrial H_2_O_2_ has been accepted as an indicator of mitochondrial ROS production [19]. Superoxide radicals formed in the electron transport chain (ETC) are immediately converted to H_2_O_2_ (catalyzed by superoxide dismutase), which diffuses across the mitochondrial membranes and serves as a stoichiometric indicator of ETC ROS production. Isolated mitochondria (250 µg protein) were resuspended in buffer, and ETC stimulation was performed with substrates of complex I (pyruvate/malate/glutamate, 5 mM each) or complex II (succinate, 5 mM). Two inhibitors of ETC were used: rotenone (6.25 µM for inhibition of complex I) and antimycin A (1 µM for inhibition of complex III). The concentration of H_2_O_2_ was determined by using the amplex red (10 µM) in the presence of horseradish peroxidase (0.6 units). Excitation and emission wavelengths of amplex red were fixed at 530 and 590 nm, respectively.

### 4.3. Statistics

All variables were tested for normality using the Shapiro–Wilk test. Results are expressed as mean ± the standard error of the mean (SEM). Statistical comparisons of means were performed using the analysis of variance (ANOVA) followed by a Bonferroni post hoc test for multiple comparisons. A *p*-value of less than 0.05 was considered statistically significant.

## 5. Conclusions

We demonstrated that in our rodent model, CsA significantly reduced myocardial ischemia reperfusion injuries even when administered 10– 20 min after the onset of reperfusion. CsA-mediated intervention may offer a reliable alternative strategy to ischemic postconditioning to protect the ischemic heart, and this finding may be relevant for heart grafts harvested from DCD donors.

## Figures and Tables

**Figure 1 ijms-23-12858-f001:**
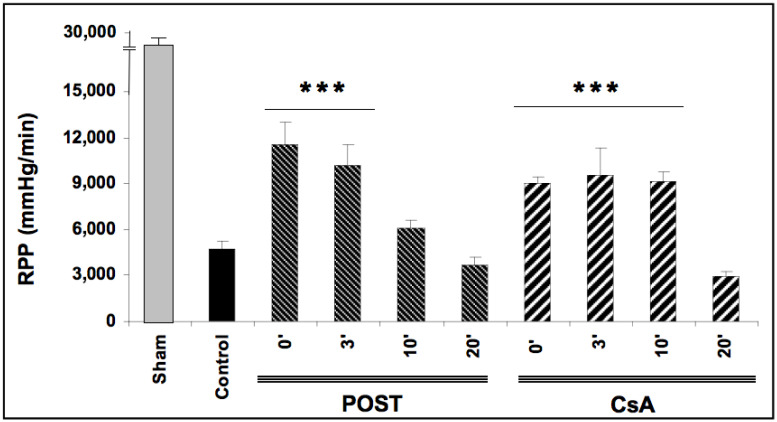
Functional recovery (rate pressure product or RPP) assessed at reperfusion. Postconditioning (POST) lost its efficacy if delayed beyond 3 min of reperfusion, whereas cyclosporine A (CsA) administration showed higher RPP even after a 10 min delay. N = 6 hearts/group. *** *p* < 0.001 vs. control.

**Figure 2 ijms-23-12858-f002:**
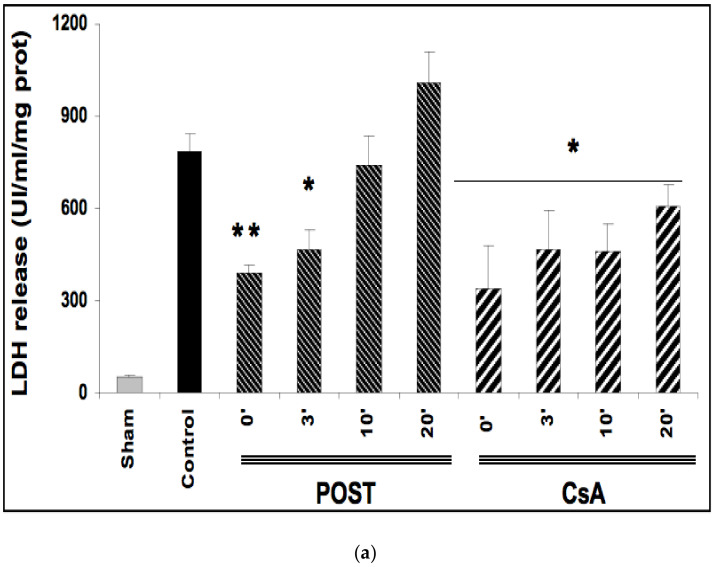
(**a**)**:** Myocardial necrosis assessed by LDH release (Panel **a**) and infarct size measurement (Panel **b**). Postconditioning (POST) lost its efficacy if delayed beyond 3 min of reperfusion, whereas CsA reperfusion showed higher RPP even after 10 min of reperfusion. N = 6 hearts/group. * *p* < 0.05, ** *p* < 0.01 vs. control. (**b**): Myocardial necrosis assessed by LDH release (Panel **a**) and infarct size measurement (Panel **b**). Postconditioning (POST) lost its efficacy if delayed beyond 3 min of reperfusion, whereas CsA reperfusion showed higher RPP even after 10 min of reperfusion. N = 6 hearts/group. ** *p* < 0.01 vs. control. (**c**): Example of heart slices of the left ventricle after the sequence of ischemia reperfusion and triphenyltetrazolium chloride staining. The colored areas (brick red) correspond to the viable tissue, while the clear areas (whitish) are the dead areas resulting from the infarction. Infarct size is expressed as % of infarcted areas compared to healthy areas.

**Figure 3 ijms-23-12858-f003:**
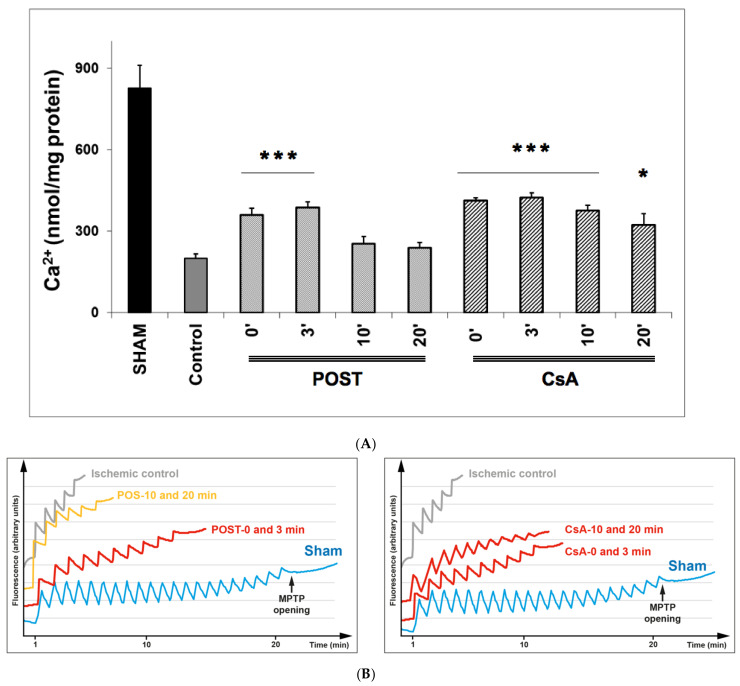
(**A**) Calcium retention capacity (CRC) is used as an indicator of mitochondrial permeability transition pore (MPTP), with the presence of complex II substrates. Delayed protection maintained higher CRC in all CsA groups, while MPTP inhibition was abolished in delayed POST-10 and -20. The insert shows classical plots obtained on sham, ischemic control, and protected hearts. A schematic diagram is shown in Appendix A to better understand the principle of CRC measurement. Abbreviations: CsA for cyclosporine A and POST for ischemic postconditioning. N = 6 hearts/group. * *p* < 0.05 and *** *p* < 0.001 vs. control. (**B**): Example of dot plot of CRC measurements obtained from each group of hearts. Pulses of 10 nmol of Ca2+ per mg of protein were added every minute to the populations of mitochondria. The calcium pulse was recorded as a fluorescence peak, and the Ca2+ was then rapidly taken up by the mitochondrial calcium uniporter. When the MPTP was open, the mitochondria and a release of Ca2+ were observed. The number of pulses was used as an indicator of the susceptibility of MPTP to mitochondrial Ca2+ overload.

**Figure 4 ijms-23-12858-f004:**
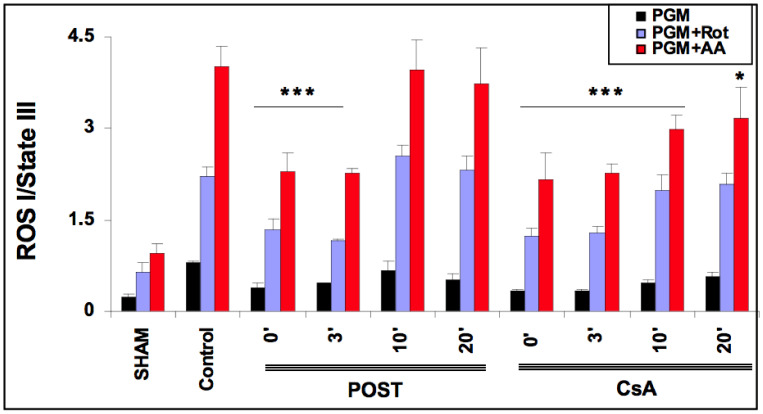
Extramitochondrial H_2_O_2_ index assessed in the presence of pyruvate-glutamate-malate- (PGM), substrates of complex I, PGM + rotenone (Rot) and PGM + rotenone + antimycin (AA). 20 min after the onset of reperfusion, the H_2_O_2_ index remained lower in the delayed CsA group but returned to control values in delayed POST-10 and -20. Abbreviations: CsA for cyclosporine A and POST for ischemic postconditioning. N = 6 hearts/group. * *p* < 0.05 and *** *p* < 0.001 vs. control.

**Figure 5 ijms-23-12858-f005:**
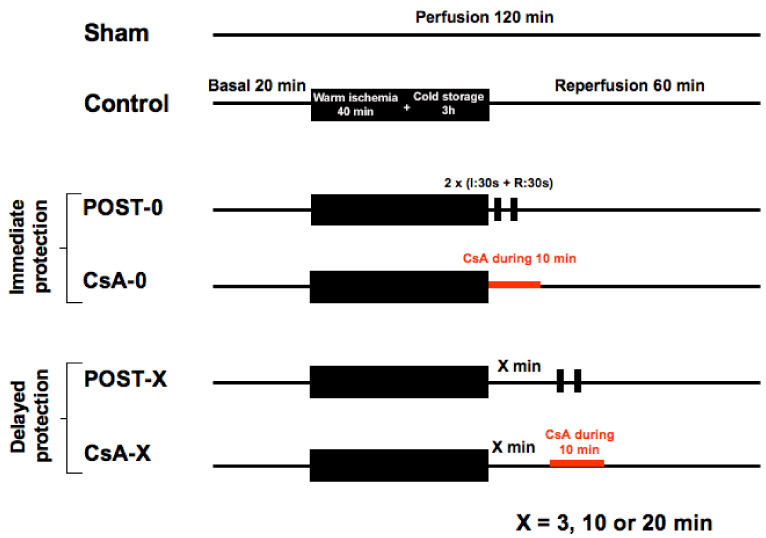
The experimental design included 10 groups of 12 animals. Sham underwent no ischemia reperfusion. All other groups experienced 40 min of ischemia followed by 60 min of reperfusion. POST-0 group (postconditioning without delay) consisted of two sequences of ischemia reperfusion at the onset of reperfusion. CsA-0 group (cyclosporin A at reperfusion without delay) involving CsA perfusion at 250 nM during the first 10 min of reperfusion. CsA and POST interventions were deferred for delayed groups by 3, 10, or 20 min.

**Table 1 ijms-23-12858-t001:** Functional recovery assessed at reperfusion. Most functional data showed higher recovery in CsA groups until 10 min of delayed intervention.

Groups	RPP 60’ (mmHg/min)	PVGD	HR (beats/min)	LVSP	LVEDP	Dp/dt max (mmHg/s)	Dp/dt min (mmHg/s)	CF (mL)
**SHAM**	26,721 ± 164	97 ± 4	303 ± 2	95 ± 5	3.41 ± 3	1815 ± 12	1492 ± 63	766 ± 81
**Control**	4765 ± 508	8.33 ± 1	296 ± 16	81 ± 6	72.67 ± 6	136 ± 20	96 ± 18	446 ± 17
**POST**								
0’	11,501 ± 1507 ***	23.83 ± 2 ***	303 ± 9	95 ± 5	72 ± 4	418 ± 25 ***	334 ± 24 ***	456 ± 42
3’	10,127 ± 1374 ***	22.16 ± 3 ***	318 ± 2	80 ± 8	58.67 ± 9 *	391 ± 45 ***	323 ± 42 ***	406 ± 47
10’	6125 ± 475	10 ± 1	310 ± 6	84 ± 6	74 ± 6	163 ± 25	131 ± 13	364 ± 18
20’	3649 ± 527	6 ± 1	296 ± 11	87 ± 4	80.83 ± 5	110 ± 21	78 ± 17	374 ± 55
**CsA**								
0’	9084 ± 341 ***	18.5 ± 3 ***	284 ± 20	91 ± 7	73 ± 6	325 ± 43 ***	246 ± 31 ***	354 ± 28
3’	9533 ± 1827 ***	21.57 ± 4 ***	286 ± 9	90 ± 10	67.42 ± 7	412 ± 76 ***	326 ± 67 ***	461 ± 37
10’	9099 ± 696 ***	16.44 ± 1 ***	306 ± 4	86 ± 4	70.70 ± 4	297 ± 26 ***	232 ± 21 ***	366 ± 24
20’	3965 ± 269	5.5 ± 0.5	299 ± 1	73 ± 3	66 ± 3	98 ± 12	72 ± 8	368 ± 48

Abbreviations: CsA for cyclosporine A, POST for ischemic postconditioning, HR for heart rate, LVSP for left ventricular systolic pressure, LVEDP for left ventricular end-diastolic pressure, RPP for rate pressure product, dP/dt max for maximum rate of rise of the LV pressure, dP/dt min for maximum isovolumetric rate of relaxation and CF for coronary flow. N = 6 hearts/group. * *p* < 0.05 and *** *p* < 0.001 vs. control.

**Table 2 ijms-23-12858-t002:** Mitochondrial oxygen consumption evaluated by measuring state III, state IV, and RCI (respiratory control index = state III/state IV). Higher RCI was maintained in the CsA group until 10 min of delayed intervention. Abbreviations: CsA for cyclosporine A, POST for ischemic postconditioning. N = 6 hearts/group. * *p* < 0.05. ** *p <* 0.01 vs control.

Groups	State III (nmol O_2_/min/mg prot)	State IV (nmol O_2_/min/mg prot)	RCI (State III/State IV)
**SHAM**	101.14 ± 10.85	19.62 ± 2.49	5.23 ± 0.17
**Control**	30.89 ± 4.27	18.86 ± 2.21	1.63 ± 0.04
**POST**			
0’	48.75 ± 7.66 **	23.10 ± 3.28	2.10 ± 0.09 *
3’	44.24 ± 0.85 *	21.55 ± 1.07	2.07 ± 0.11
10’	35.59 ± 4.41	23.38 ± 5.59	1.64 ± 0.12
20’	36.88 ± 3.41	23.19 ± 2.24	1.65 ± 0.04
**CsA**			
0’	50.53 ± 6.88 **	21.04 ± 0.72	2.39 ± 0.30 **
3’	48.26 ± 4.23 **	21.82 ± 0.97	2.21 ± 0.14 *
10’	42.84 ± 3.11 *	20.59 ± 1.50	2.08 ± 0.03 *
20’	38.36 ± 1.26	22.90 ± 0.66	1.67 ± 0.02

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
