# Peer review of "Postconditioning by Delayed Administration of Ciclosporin A: Implication for Donation after Circulatory Death (DCD)"

_ijms, 2022, doi:10.3390/ijms232112858_

Round 1

Reviewer 1 Report

In the present work, author determined whether protecting the heart during reperfusion also known as Postconditioning (ischemic or with ciclosporin-A (CsA)) reduces myocardial ischemia-reperfusion injury in a model of cardiac arrest when applied either at the onset of reperfusion or during reperfusion.  Based on results on the cardiac functional recovery, myocardial infarct size and mitochondrial calcium retention capacity this study provided an evidence of the efficacy of MPTP inhibition as protective strategy for DCD heart transplantation, even after delay intervention at reperfusion.  However, the authors should consider minor revisions prior publication as listed below:

 1. Addition of a group that will receive ischemic post + CsA at the efficient protocol would confirm whether these two effect on the CRC is additive or not. Addition of this group would strengthen the study.

 2. The reference list formatting needs to be fixed. 

Author Response

I am very grateful to Reviewer-1 for the favorable rating of the paper and his/her constructive remarks. Regarding point-1, it is indeed very important to ascertain that the cardioprotection methods used are not additive. To answer this question, we had previously carried out work in this direction, which had shown that, in our model, the 2 cardioprotective methods used simultaneously ("POST + CsA") were not additive. The results of this preliminary work have been presented in figure D of the "Supplementary Data". This result is not surprising since other teams had already shown that the addition of 2 methods (eg: Preconditioning + Postconditioning) were not additive. As all these methods use the same mechanisms of action (with a final effector, the MPTP), this seems coherent to us. However, these data do not explain why some methods are effective even after a delay in application at the reperfusion, as is the case in our work with CsA.

Regarding point-2, the format of bibliographical references has been harmonized.

Reviewer 2 Report

The presented study highlights the effects of deferred postconditioning (POST) of ischemic rat hearts. The main conclusion reached by the authors is that the window of maneuver the ischemic heart is very narrow, in order to prevent irreversible damage. This conclusion has been reached by other independent studies. The novelty of this study relies on the observations that irreversible damage to the ischemic heart caused by reperfusion could be delayed (prevented) by application of CsA. These findings suggest that mitochondrial activity (and integrity) must be preserved as a way to protect the heart from reperfusion injury. Treatment with CsA, might be useful in widening the window for maneuvering the ischemic hearts intended for transplantation in humans, reducing the risk of tissue damage. This is a sound idea, in principle. However, the manuscript is presented in a way that results confusing for the reader to catch up a clear and simple message. My comments and suggestions can be accessed here in the attached PDF file. In summary:

1. Authors should provide exemplars of infarct size for all the groups. 

2. Exemplars of CRC for all the groups.

3. Real "p" values.

4. Data not shown, has to be show as a supporting figure. 

5. The conclusions have to be attained to the restrictions of the model employed in the study, and/or be sustained with appropriate references of studies with larger animals. 

 6. One of the most intriguing observations was that ROS production (H2O2) is reduced in hearts with delayed CsA. The authors do not discuss this in detail in the manuscript, but I would like to know what their interpretation is.

Author Response

I thank you very much for the great attention that the Reviewer-2 has given to the reading of our manuscript.

We have therefore provided an answer to the various questions, comments and suggestions proposed by the Reviewer-2. We used the file containing the remarks of Reviewer-2 on which we directly answered point by point. In addition, the changes made also appear in the new version of the manuscript of the article (areas highlighted in yellow). Only the "p" values are still pending for the final version. In order to better complete our answers, we have also added additional results (see Figures A, B and C) in the new "Supplementary Data" file.

Round 2

Reviewer 2 Report

Thank you for the answers provided. There are minor comments from the reviewer mostly on the CRC data. This reviewer suggested to include exemplars of the actual experiments performed under the conditions summarized in the bars plot (Fig. 4). The added diagram is useful to understand the principle of CRC, however I did request exemplars of real experiments for each condition tested.  P values are missing.

Author Response

As suggested by the Reviewer-2, the dot plots of the MPTP opening measurement corresponding to each group studied are now shown in Figure 4 Panel-B. To facilitate reading, 2 graphs have been produced, one presenting the plots of the POST groups and the other presenting the CsA groups, in comparison with the Sham and Control groups.  

As proposed by the Reviewer-2, the exact "p" values are now shown when 2 groups with a single condition are compared. When the text compares several groups and/or several conditions, it seemed to us simpler and more readable to use the significance thresholds (p<0.05, 0.01 and 0.001), in order to group the useful information.

We thank the Reviewer-2 for his understanding and for the important work he has devoted to improving our paper.